# Association between cerebral oximetry and return of spontaneous circulation following cardiac arrest: A systematic review and meta-analysis

Yupeng Liu⬥, Kunpeng Jing, Hongwei Liu, Yongfang Mu, Zhaoqin Jiang, Yadong Nie, Chongyang Zhang*

Department of Emergency, Qinhuangdao First Hospital, Hebei Province, People's Republic of China

* Zhangchongyang2006@163.com

**Data Availability Statement:** All relevant data are within the manuscript.

**Funding:** The author(s) received no specific funding for this work.

## Abstract

The present meta-analysis was based on the available studies to determine the potential role of the initial and regional cerebral oxygen saturation (rSO2) in monitoring the efficiency of cardiopulmonary resuscitation (CPR) and predicting the return of spontaneous circulation (ROSC). Three electronic databases of PubMed, Embase, and the Cochrane Library were searched to identify the studies that investigated the role of rSO2 on ROSC in CA patients throughout May 2018. The weighted mean difference (WMD) with 95% confidence interval (CI) was calculated to estimate the pooled effect using a random-effects model. Sensitivity, subgroup analyses, and publication bias were conducted. A total of 13 studies involving 678 CA patients (300 in-hospital (IH) patients, and 378 out-hospital (OH) patients) were included. The summary WMD suggested that ROSC patients were associated with higher initial rSO2 (WMD: 10.10%; 95% CI: 5.66–14.55; P<0.001) and mean rSO2 (WMD: 14.16%; 95% CI: 10.51–17.81; P<0.001) levels during CA and ROSC as compared to the non-ROSC. The results of meta-regression suggested that the male percentage and the location of cardiac arrest might bias the initial or mean rSO2 and the incidence of ROSC. These significant differences were observed in nearly all subsets. The findings of this study suggested that high initial or mean rSO2 levels were both associated with an increased incidence of ROSC in CA patients undergoing CPR. These correlations might be affected by the percentage of males or the location of cardiac arrest, thereby necessitating further large-scale studies to substantiate whether these correlations differ according to gender and the location of cardiac arrest.

## Introduction

Cardiac arrest (CA) is an emergency endpoint of many diseases and conditions with high variability among individuals and is affected by genetic factors, age, environment, and gender [1]. The rate of survival to hospital discharge after CA was 6.3% in 2016 [2]. The Emergency Medical Services improved the survival rate to 12%, while the prognosis of the surviving patients

**Competing interests:** The authors have declared that no competing interests exist.

with a satisfactory neurological function was <10% [3]. The high mortality rate of CA patients due to hypoxic-ischemic brain injury (HIBI), resulted in neuronal death and diffused brain edema [4–7]. The withdrawal of life-sustaining treatment in patients was attributed to the prognostication of a poor neurological outcome [8,9]. Therefore, effective markers are an urgent requirement to predict the prognosis of CA patients undergoing cardiopulmonary resuscitation (CPR).

Near-infrared spectroscopy is a non-invasive optical technique that is widely used to measure the regional cerebral oxygen saturation (rSO2) in the superficial parts of the frontal lobe [10]. rSO2 provides real-time information on oxygen delivery during resuscitation in CA patients [11]. rSO2 is used for patients undergoing neonatal intensive care and has been introduced in in-hospital (IH) and out-hospital (OH) CA patients, but its role in monitoring the effectiveness of CPR has not yet been established [12–14]. A previous meta-analysis based on nine studies found that the return of spontaneous circulation (ROSC) in CA patients is associated with high initial and average rSO2 [15]. Whether these correlations differed among patients with different characteristics is not yet illustrated [15]. Therefore, we attempted a large-scale systematic review and meta-analysis of the available studies to illustrate the associations of rSO2 with ROSC in CA patients and to compare these correlations according to the patients' characteristics.

## Methods

### Data sources, search strategy, and selection criteria

This study was conducted, and results reported according to the protocol for the meta-analysis of observational studies in epidemiology [16]. The studies published in the English language, without any restriction on publication status, demonstrated that the association of rSO2 levels and the incidence of ROSC was eligible for inclusion in this meta-analysis. We systematically searched the electronic databases, such as PubMed, Embase, and the Cochrane Library for studies published from inception to May 2018 (print date), and the core search terms were "cerebral oximetry" AND ("cardiac" OR "cardiopulmonary") AND ("arrest" OR "resuscitation"). The reference lists from the potentially included studies were also searched to select additional eligible studies. The study selection criteria based on the Patient, Intervention, Comparison, Outcome, and study (PICOS) standard were employed to identify the eligible studies.

Two authors independently performed the literature search and selected the eligible studies using a standard flow. Any disagreement between the two authors was resolved by a group discussion to reach a consensus. The study inclusion criteria are listed as follows: (1) Participants: all the patients with CA and undergoing CPR; (2) Exposure: ROSC CA patients; (3) Control: non-ROSC patients; (4) Outcome: rSO2 was measured using near-infrared spectroscopy technique; (5) Study design: prospective or retrospective.

### Data collection and quality assessment

The data were collected and quality assessed by two authors independently, and any inconsistencies were resolved by the corresponding author referring to the original article. The data items collected included the first author's last name, publication year, study design, country, sample size, mean age, the percentage of males, number of IH/OH, assessment of exposure, duration of resuscitation, and investigated outcomes. The Newcastle–Ottawa Scale (NOS) was used to comprehensively assess the methodological quality of observational studies in the meta-analysis [17]. The NOS was based on selection (4 items), comparability (1 item), and outcome (3 items). The "star system" of NOS ranged from 0–9, and the study that scored ≥7 was regarded as high quality.

## Statistical analysis

The correlations between initial or mean rSO2 and incidence of ROSC were based on the mean, standard deviation, and sample size in each group in the individual study. The pooled outcomes were calculated using a random-effects model for ROSC *vs*. non-ROSC patients [18,19]. The heterogeneity among the included studies was calculated using $I^2$, and 25%, 50%, or 75% was considered as low, moderate, and high heterogeneity, respectively [20,21]. The sensitivity analysis evaluated the impact of a single study from the overall analysis [22]. Subgroup analyses and meta-regression were performed to determine whether the characteristics of the studies and populations of patients influenced the conclusion of the meta-analysis [23]. The subgroup analyses were based on the publication year (this could be a confounder because of the changes in practice and guidelines over time), study design (prospective trials have more power than retrospective studies), mean age (age might influence cerebral oximetry and the outcomes of cardiac arrest), sex (males usually have worst cardiac outcomes than females), the location of cardiac arrest (cardiac arrests occurring at the hospital will be managed more promptly than those occurring outside), and study quality (study and data quality might affect the quality of the analyses and conclusions). Funnel plots and Egger and Begg tests' results were employed to evaluate any potential publication biases [24,25]. The reported P-values are two-sided, and P<0.05 was considered as statistically significant among the included studies. All the statistical analyses were conducted by STATA software (version 10.0; Stata Corporation, College Station, TX, USA).

## Results

### Literature search

The electronic search retrieved 201 articles: 72 from PubMed, 124 from Embase, and five from the Cochrane library database. Subsequently, 162 studies were excluded as they were duplicates or irrelevant topics. As a result, a total of 39 potentially eligible studies were selected, and after detailed evaluation, 19 were excluded due to insufficient data, four studies presented other interventions, and three studies evaluated other topics. Finally, 13 studies were selected for the final analysis [26–38]. The study selection process is illustrated in Fig 1, and the baseline characteristics of the included studies are shown in Table 1.

### Study characteristics

A total of 13 studies involving a total of 678 CA patients (300 IH patients and 378 OH patients) were included in this meta-analysis. Of these, 10 had a prospective design, and the remaining three studies were retrospective. The mean age of the patients was 64.8–79.5 years, and 10–183 patients were included in each study. Eleven studies were conducted in Western countries, and the remaining two studies were conducted in Eastern countries. The data on the correlation between initial rSO2 and ROSC were available from 7 studies and that between mean rSO2 and ROSC were available from 10 studies. The study quality was evaluated using NOS; seven of the included studies scored 7, four studies scored 6, and the remaining two studies scored 5.

### Initial rSO2

The correlation established between initial rSO2 and ROSC from 7 pooled studies demonstrated that ROSC in patients was associated with high initial rSO2 level (weighted mean difference (WMD): 10.10%; 95% confidence interval (CI): 5.66–14.55; P<0.001; Fig 2), and significant heterogeneity was observed across the included studies ($I^2$: 72.1%; P = 0.001). The

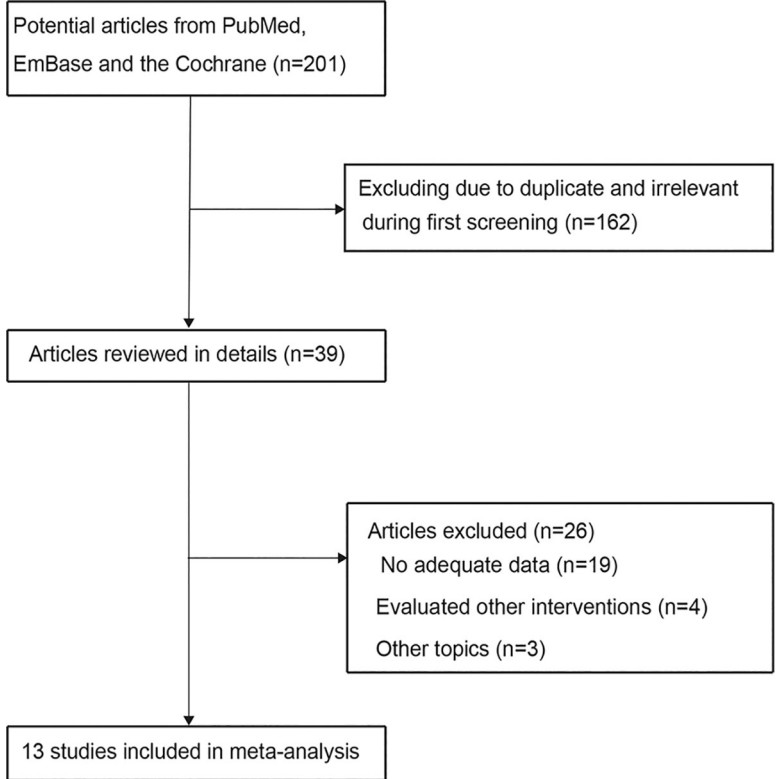

**Fig 1. Study selection process.**

conclusion from the sensitivity analysis was not altered after the sequential exclusion of individual studies from the overall analysis (Table 2). Meta-regression analyses indicated that the proportion of males (P = 0.032) and the location of cardiac arrest (P<0.001) could influence the association of initial rSO2 and ROSC. Furthermore, the subgroup analysis did not establish a significant correlation between initial rSO2 and ROSC if the study included both IH and OH CA patients, while significant associations were noted in all the other subsets (Table 3).

## Mean rSO2

The summary results of a total of 10 studies were pooled to analyze the correlation between mean rSO2 and ROSC, and WMD indicated that ROSC was associated with greater mean rSO2 levels as compared to the non-ROSC (WMD: 14.16%; 95% CI: 10.51–17.81; P<0.001; Fig 3). Although substantial heterogeneity across the included studies was detected ($I^2$: 75.6%; P<0.001), the conclusion was not affected by the sequential exclusion of each study from the overall analysis (Table 2). The results of meta-regression analyses indicated that the percentage of males (P = 0.002) and the location of cardiac arrest (P = 0.001) significantly influenced the correlation between median rSO2 and ROSC. Also, the subgroup analysis indicated a significant association among all the subsets based on pre-defined factors (Table 3).

## Publication bias

The funnel plots were reviewed, and the potential for publication bias for initial (Fig 4A) and mean rSO2 levels (Fig 4B) was not excluded. The Egger and Begg tests results did not show

**Table 1. Baseline characteristics of included studies and patients.**

| First author's surname | Study design | Country | Sample size | Mean age (years) | Percentage male (%) | Number of IH/OH | Assessment of exposure | Duration of resuscitation (minutes) | Reported outcomes | Study quality |
|---|---|---|---|---|---|---|---|---|---|---|
| Ahn 2013 [25] | Prospective | USA | 50 | 64.8 | 72.0 | 36/14 | Equanox 7600, Nonin Medical, Inc., Plymouth, MN, USA | 34.4 | Initial rSO2%; mean rSO2% | 7 |
| Parnia 2014 [26] | Retrospective | USA | 34 | 71.0 | 64.7 | 34/0 | Equanox, Nonin, Plymouth, MI, USA INVOS, Covidien, Mansfield, MA, USA | 19.0 | Mean rSO2% | 6 |
| Genbrugge 2015 [27] | Prospective | Belgium | 49 | 73.0 | 63.3 | 0/49 | Equanox Advance, Nonin Medical, Inc., Plymouth, MN, USA | 21.0 | Initial rSO2%; mean rSO2% | 7 |
| Singer 2014 [28] | Retrospective | USA | 59 | 68.7 | 84.7 | 0/59 | Equanox, Nonin, Plymouth MI, USA | NA | Mean rSO2% | 6 |
| Fukuda 2014 [29] | Prospective | Japan | 69 | 66.1 | 69.6 | 0/69 | INVOS 5100, Covidien, Boulder, CO, USA | 38.7 | Initial rSO2% | 7 |
| Koyama 2013 [30] | Prospective | Japan | 15 | 79.5 | 66.7 | 0/15 | Hamamatsu Photonics, Hamamatsu-Shi, Shizuoka, Japan | NA | Initial rSO2% | 6 |
| Parnia 2012 [31] | Retrospective | USA | 15 | 73.8 | NA | 15/0 | INVOS Somanetics, Troy, USA | 16.3 | Mean rSO2% | 5 |
| Meex 2013 [32] | Prospective | Belgium | 14 | 66.0 | 74.4 | 5/9 | FORE-SIGHT (CAS Medical Systems, Branford, CT, USA) and Equanox Advance (Nonin Medical, Inc., Plymouth, MN, USA) | 25.0 | Initial rSO2% | 6 |
| Schewe 2014 [33] | Prospective | Germany | 10 | 73.0 | 80.0 | 0/10 | Equanox 7600, Nonin Medical, Inc., Plymouth, MN, USA | NA | Mean rSO2% | 5 |
| Ibrahim 2015 [34] | Prospective | USA | 27 | 65.6 | 96.3 | 27/0 | Invos 5100 C near infrared spectroscopy device (Invos Somanetics, Troy, USA) | 20.8 | Initial rSO2%; mean rSO2% | 7 |
| Parnia 2016 [35] | Prospective | USA and United Kingdom | 183 | 68.6 | 60.7 | 183/0 | Equanox 7600, Nonin Medical, Plymouth, MN, USA | 28.7 | Mean rSO2% | 7 |
| Prosen 2018 [36] | Prospective | Slovenia | 53 | 68.5 | 84.9 | 0/53 | INVOS oximeter (Somanetics Corporation, Troy, MI, USA) | 16.2 | Initial rSO2%; mean rSO2% | 7 |
| Singer 2018 [37] | Prospective | USA | 100 | 69.0 | 73.0 | 0/100 | Equanox 7600, Nonin Medical, Plymouth, MN, USA | NA | Mean rSO2% | 7 |

*IH: in-hospital arrests; NA: not available; OH: out-of-hospital arrests; rSO2: regional cerebral oxygen saturation.

any evidence of publication bias for initial rSO2 (P-value for Egger: 0.258; P-value for Begg: 0.764) and mean rSO2 levels (P-value for Egger: 0.827; P-value for Begg: 0.210).

## Discussion

The survival rate of CA patients is low, and hence, the prediction role of initial and mean rSO2 should be explored. Numerous studies have demonstrated the positive association of initial and mean rSO2; however, whether these correlations differ according to the study (publication

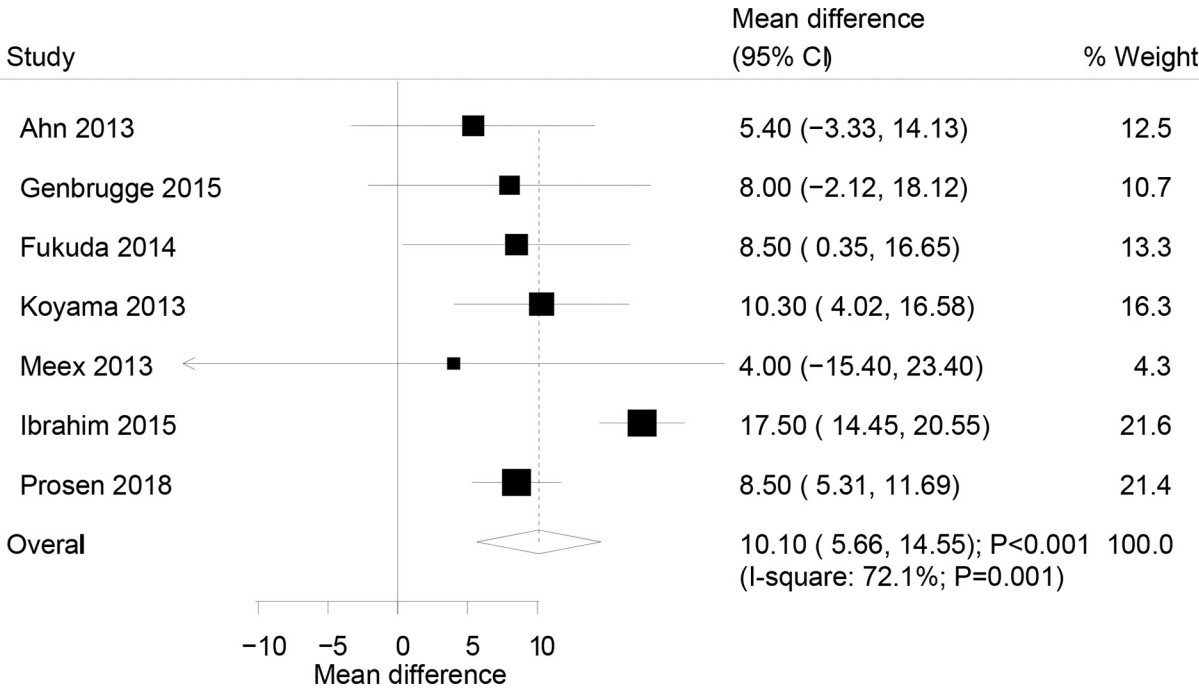

**Fig 2. Association of the initial rSO2 level with the incidence of the return of spontaneous circulation.**

year, study design, and study quality) or patients' (mean age, sex, and location of cardiac arrest) characteristics are yet to be elucidated. In this comprehensive quantitative meta-analysis, 678 CA patients from 13 studies with a broad range of characteristics were included. The summary results indicated that ROSC *vs*. non-ROSC was significantly associated with high

**Table 2. Sensitivity analyses for initial rSO2 and mean rSO2.**

| Outcomes | Excluding study | WMD and 95% CI | P-value | Heterogeneity (%) | P-value for heterogeneity |
|---|---|---|---|---|---|
| Initial rSO2 (%) | Ahn 2013 | 10.77 (6.02 to 15.52) | <0.001 | 74.0 | 0.002 |
| | Genbrugge 2015 | 10.30 (5.44 to 15.16) | <0.001 | 76.1 | 0.001 |
| | Fukuda 2014 | 10.27 (5.29 to 15.25) | <0.001 | 75.9 | 0.001 |
| | Koyama 2013 | 9.91 (4.66 to 15.16) | <0.001 | 76.4 | 0.001 |
| | Meex 2013 | 10.36 (5.75 to 14.97) | <0.001 | 76.1 | 0.001 |
| | Ibrahim 2015 | 8.43 (5.96 to 10.90) | <0.001 | 0.0 | 0.961 |
| | Prosen 2018 | 10.44 (5.24 to 15.64) | <0.001 | 64.9 | 0.014 |
| Mean rSO2 (%) | Ahn 2013 | 14.02 (10.00 to 18.04) | <0.001 | 78.2 | <0.001 |
| | Parnia 2014 | 13.38 (9.70 to 17.06) | <0.001 | 75.7 | <0.001 |
| | Genbrugge 2015 | 14.65 (10.53 to 18.77) | <0.001 | 76.1 | <0.001 |
| | Singer 2014 | 14.79 (10.85 to 18.73) | <0.001 | 75.7 | <0.001 |
| | Parnia 2012 | 13.94 (10.06 to 17.82) | <0.001 | 78.1 | <0.001 |
| | Schewe 2014 | 14.40 (10.68 to 18.13) | <0.001 | 77.9 | <0.001 |
| | Ibrahim 2015 | 12.31 (9.83 to 14.78) | <0.001 | 31.4 | 0.167 |
| | Parnia 2016 | 14.68 (10.56 to 18.79) | <0.001 | 75.2 | <0.001 |
| | Prosen 2018 | 13.98 (10.06 to 17.91) | <0.001 | 78.2 | <0.001 |
| | Singer 2018 | 14.79 (10.87 to 18.71) | <0.001 | 75.9 | <0.001 |

*rSO2: regional cerebral oxygen saturation; WMD: weighted mean difference.

**Table 3. Subgroup analyses for initial rSO2 and mean rSO2.**

| Outcomes | Subgroup | WMD and 95% CI | P-value | Heterogeneity (%) | P-value for Meta-regression |
|---|---|---|---|---|---|
| Initial rSO2 (%) | | **Publication year** | | | |
| | 2015 or after | 11.86 (4.58 to 19.14) | 0.001 | 88.2 | 0.057 |
| | 2015 previous | 8.38 (4.16 to 12.59) | <0.001 | 0.0 | |
| | | **Study design** | | | |
| | Prospective | 10.10 (5.66 to 14.55) | <0.001 | 72.1 | - |
| | Retrospective | - | - | - | |
| | | **Mean age (years)** | | | |
| | ≥ 70.0 | 9.66 (4.33 to 15.00) | <0.001 | 0.0 | 0.354 |
| | < 70.0 | 10.14 (4.27 to 16.01) | 0.001 | 80.5 | |
| | | **Percentage male (%)** | | | |
| | ≥ 80.0 | 13.01 (4.19 to 21.83) | 0.004 | 93.7 | 0.032 |
| | < 80.0 | 8.32 (4.43 to 12.21) | <0.001 | 0.0 | |
| | | **Location of cardiac arrest** | | | |
| | IH | 17.50 (14.45 to 20.55) | <0.001 | - | <0.001 |
| | OH | 8.78 (6.18 to 11.37) | <0.001 | 0.0 | |
| | Both | 5.16 (-2.80 to 13.13) | 0.204 | 0.0 | |
| | | **Study quality** | | | |
| | High | 10.25 (4.76 to 15.75) | <0.001 | 80.5 | 0.423 |
| | Low | 9.70 (3.73 to 15.67) | 0.001 | 0.0 | |
| Mean rSO2 (%) | | **Publication year** | | | |
| | 2015 or after | 13.67 (8.53 to 18.82) | <0.001 | 85.9 | 0.632 |
| | 2015 previous | 14.84 (9.29 to 20.39) | <0.001 | 52.1 | |
| | | **Study design** | | | |
| | Prospective | 13.62 (9.28 to 17.96) | <0.001 | 79.4 | 0.556 |
| | Retrospective | 16.17 (7.12 to 25.21) | <0.001 | 72.9 | |
| | | **Mean age (years)** | | | |
| | ≥ 70.0 | 15.19 (8.03 to 22.35) | <0.001 | 57.8 | 0.377 |
| | < 70.0 | 13.74 (9.03 to 18.46) | <0.001 | 82.7 | |
| | | **Percentage male (%)** | | | |
| | ≥ 80.0 | 14.68 (7.07 to 22.28) | <0.001 | 81.7 | 0.002 |
| | < 80.0 | 12.55 (9.16 to 15.95) | <0.001 | 52.7 | |
| | | **Location of cardiac arrest** | | | |
| | IH | 17.81 (10.97 to 24.66) | <0.001 | 85.0 | 0.001 |
| | OH | 10.63 (8.13 to 13.13) | <0.001 | 0.0 | |
| | Both | 15.50 (8.93 to 22.07) | <0.001 | - | |
| | | **Study quality** | | | |
| | High | 13.94 (9.48 to 18.41) | <0.001 | 82.4 | 0.444 |
| | Low | 14.83 (6.92 to 22.74) | <0.001 | 62.1 | |

*IH: in-hospital arrests; OH: out-of-hospital arrests; rSO2: regional cerebral oxygen saturation; WMD: weighted mean difference.

initial and mean rSO2 levels. Furthermore, these correlations might be influenced by the percentage of males and the location of cardiac arrest. Finally, a significant association was noted in all the subsets except for the study that included both IH and OH CA patients.

The summary result indicated ROSC patients with high initial rSO2 level, while 3/7 studies reported inconsistent results. Ahn et al. demonstrated that a critical role of cerebral oximetry on ROSC in asystole and pulseless electrical activity patients, while no significant association occurred in shockable patients [26]. The study pointed out that ROSC could not achieve

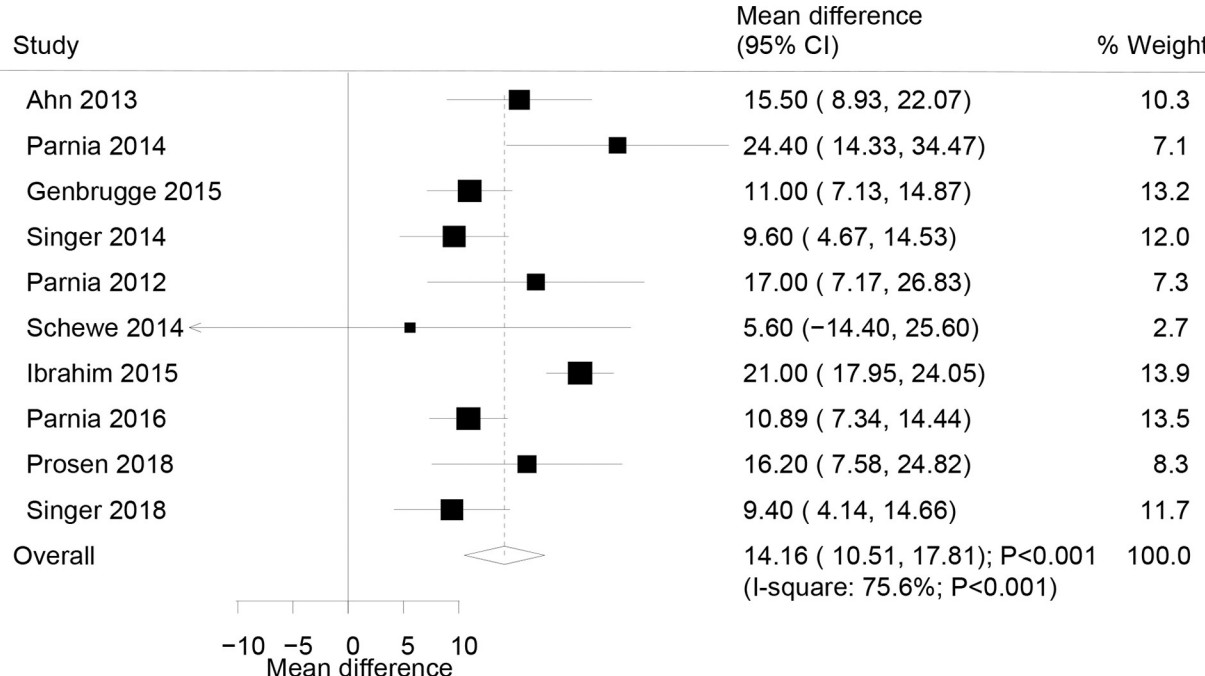

**Fig 3. Association of the mean rSO2 level with the incidence of the return of spontaneous circulation.**

successful defibrillation or chemical cardioversion in patients with refractory shock [39,40]. Genbrugge et al. did not establish a significant correlation between ROSC and initial rSO2 level (P = 0.07) [28], and this phenomenon could be interpreted as the short duration between the emergency call and the start of advanced life support associated with a high initial rSO2 value. Genbrugge et al. [41] reported the results of the Copernicus I study and showed that rSO2 could be used during pre-hospital advanced life support as a marker to predict the return of spontaneous circulation; an increase in rSO2 of at least 15% predicted the return of spontaneous circulation. Meex et al. included 14 patients and found that the initial rSO2 value could not predict the mortality [33]. Although these studies did not report any significant associations between ROSC and initial rSO2 values, an increase in the rSO2 values in ROSC patients was observed in all the included studies.

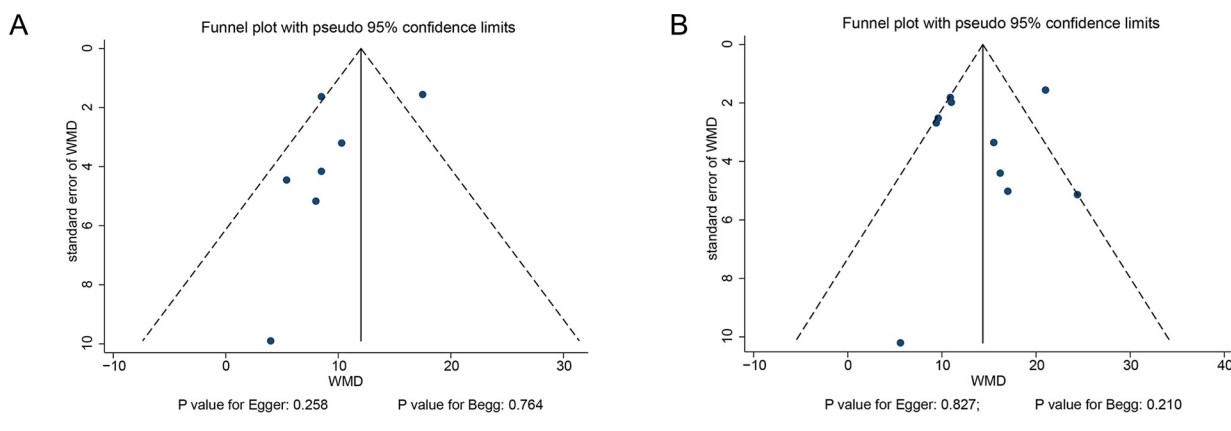

**Fig 4.** Funnel plots for initial rSO2 (A) and mean rSO2 (B) levels.

The summary result indicated a positive correlation between ROSC and mean rSO2 values, and almost all the included studies reported similar conclusions. Schewe et al. did not show a significant association between ROSC and rSO2 value, which might be attributed to a small number of patients included in the study [34]. The ROSC in patients was closely related to oxygen delivery and circulation, and thus, exhibited a significant improvement in the quality of CPR [42,43]. Furthermore, cerebral oximetry could predict ROSC, and cerebral perfusion during CPR was associated with significant improvement in brain oxygenation and the subsequent survival and neurological outcomes. Finally, the rSO2 value was correlated with the quality of CPR during CA and ROSC or survival during resuscitation [44,45].

Subgroup analyses indicated significant associations between rSO2 values and ROSC in all the subsets, while the results of meta-regression suggested that these correlations might be affected by gender and the location of cardiac arrest. The phenomenon could be ascribed to less mortality in females than males in CA survivors who received therapeutic hypothermia (P = 0.03) [46]. Furthermore, the location of cardiac arrest included IH and OH with various intervals between the emergency call and start of the advanced life support, which could affect the incidence of ROSC. Finally, these results might be variable due to the small number of studies in the corresponding subsets.

Nevertheless, the present meta-analysis has several highlights. First, the large sample size and comprehensive search provided robust results than any individual study. Second, the subgroup analyses for these associations assessed whether these correlations differed according to the publication year, study design, mean age, percentage of males, the location of cardiac arrest, and study quality. The limitations of this study were as follows: (1) several included studies had a retrospective design, which might introduce selection and recall biases; (2) potential publication bias might exist due to the meta-analysis based on the published studies; (3) this analysis was based on pooled data, and detailed calculations were not conducted as individual datasets were not available; (4) a subgroup analysis for the device used could not be performed because the number of different devices was too large and the number of studies for some devices was too small for a reliable analysis; and (5) not all studies report the same outcomes, and the reported data have to be analyzable in the context of a meta-analysis; therefore, we had to select the variables that were the most consistently reported among the included studies and had clinical meaning.

## Conclusions

Taken together, the findings of this study suggest that high initial or mean rSO2 value could predict the incidence of ROSC in CA patients. The strength of these associations was greater in males and IH patients than the corresponding subsets. However, a further large-scale prospective study should be conducted to verify the potential difference in gender and the location of cardiac arrest.

## Supporting information

**S1 Checklist.**
(DOC)

**S1 Flow diagram.**
(DOC)

## Author Contributions

**Conceptualization:** Yupeng Liu.

**Data curation:** Kunpeng Jing, Hongwei Liu, Yadong Nie.

**Formal analysis:** Yupeng Liu, Hongwei Liu.

**Methodology:** Yupeng Liu.

**Resources:** Kunpeng Jing.

**Software:** Kunpeng Jing, Zhaoqin Jiang.

**Supervision:** Yongfang Mu.

**Writing – original draft:** Yupeng Liu.

**Writing – review & editing:** Yupeng Liu, Chongyang Zhang.

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
