## [Decision Letter · Decision Letter 0]

20 Dec 2019

PONE-D-19-30622

Association between cerebral oximetry and return of spontaneous circulation following cardiac arrest: a systematic review and meta-analysis

PLOS ONE

Dear Dr Zhang,

Thank you for submitting your manuscript to PLOS ONE. After careful consideration, we feel that it has merit but does not fully meet PLOS ONE’s publication criteria as it currently stands. Therefore, we invite you to submit a revised version of the manuscript that addresses the points raised during the review process.

The reviewers and I have several issues with the study's analysis of rSO2 data. Please refer to the specific comments below.

We would appreciate receiving your revised manuscript by January 31, 2020. To enhance the reproducibility of your results, we recommend that if applicable you deposit your laboratory protocols in protocols.io, where a protocol can be assigned its own identifier (DOI) such that it can be cited independently in the future. For instructions see: http://journals.plos.org/plosone/s/submission-guidelines#loc-laboratory-protocols

We look forward to receiving your revised manuscript.

Kind regards,

Steve Lin

Academic Editor

PLOS ONE

Journal Requirements:

2. Please provide any updates you might have since the original search was performed in May 2018, or please provide the rational for ending your search at that time.

Reviewers' comments:

Reviewer's Responses to Questions

**Comments to the Author**

1. Is the manuscript technically sound, and do the data support the conclusions?

Reviewer #1: Partly

Reviewer #2: Partly

2. Has the statistical analysis been performed appropriately and rigorously? 

Reviewer #1: Yes

Reviewer #2: I Don't Know

3. Have the authors made all data underlying the findings in their manuscript fully available?

Reviewer #1: Yes

Reviewer #2: Yes

4. Is the manuscript presented in an intelligible fashion and written in standard English?

Reviewer #1: Yes

Reviewer #2: Yes

5. Review Comments to the Author

Reviewer #1: Thank you for giving me the opportunity to read this interesting manuscript. The authors were challenged to write a coherent story with the diverse data of cerebral oxygenation during cardiac arrest.

However I have a few comments.

In the abstract the authors write in the first sentence that the initial and mean rSO2 play a role in the efficiency of CPR. However it is impossible that an initial value as also the mean values, which is generated posthoc can influence the quality of CPR. The authors should adapt this sentence.

What do the authors mean with patients status in the abstract and further in the manuscript? In the Tables it is the location of the cardiac arrest. The authors should indicate this in the abstract and the first time they mention this in the manuscript.

The authors make a difference between OHCA and IHCA patients however they should also investigate the moment of measurements. As in some studies, cerebral oxygenation is measured pre-hospital, sometimes at the ED of OHCA, sometimes In hospital at the place of arrest or at the ED once the IHCA patient is brought to the ED. You can expect differences in saturation values as the time between cardiac arrest and start of measurements is different. Do the authors analyzed the data categorized following the moment of measurement OH or IH?

The authors write that cardiac arrest is a disease however they should adapt this as it is an acute endpoint of many diseases.

Line 41: the authors give a survival rate of 6,3% however they don't give any reference. Can they add a reference?

Line 52: this sentence should be adapted as none of the other studies until today could associate rSO2 and neurological outcome.

Line 53: rSO2 is not widely used pre-operative.

Line 59: what do they mean with specific characteristics?

The authors should also search under near infra red spectroscopy.

How and why was decided to use initial and mean rSO2 values?

Are the authors sure they included all manuscript about cerebral saturation and cardiac arrest during they research period? A recent study was published by Genbrugge et al. using cerebral oximetry in CA patients. Why was this study excluded? Why are the studies of Ito et al. excluded?

On what is the subgroep analysis based?

Did the authors also investigated the different devices used to measure rSO2? As some devices measure until 0 other have a higher cut off point. This can influence in the calculated mean values and initial values as some devices for example indicate 15 for all values below 15

Reviewer #2: Interesting paper, a few things for the authors to consider:

-All results should be separated based on prospective vs. retrospective and in-hospital vs out-of-hospital throughout the entire paper as it is not ideal to perform statistics combining these groups

-In your methods, describe what the term 'patient status' means, as this is a very general term

-More detail is needed in the statistical analysis section to describe what you mean by 'subgroup analysis' and 'meta-regression analysis' as these terms alone don't convey what you are specifically addressing with these analyses (i.e. what outcomes are you looking at)

-Instead of saying the relationships can be biased by 'percentage of males', it would be better to state they are influenced by gender. Additionally, if gender does influence the results, it would be beneficial to add a table that specifically separates results for males and females if this data is available.

6. PLOS authors have the option to publish the peer review history of their article (what does this mean?). If published, this will include your full peer review and any attached files.

Reviewer #1: Yes: Cornelia Genbrugge

Reviewer #2: No

---

## [Author Response · Author response to Decision Letter 0]

18 Jan 2020

Manuscript ID: PONE-D-19-30622

Title: Association between cerebral oximetry and return of spontaneous circulation following cardiac arrest: a systematic review and meta-analysis

Journal: PLOS ONE

Response to Reviewers' comments

Dear Editor, 

 We thank you for your careful consideration of our manuscript. We appreciate your response and overall positive initial feedback and made modifications to improve the manuscript. After carefully reviewing the comments made by the Reviewers, we have modified the manuscript to improve the presentation of our results and their discussion, therefore providing a complete context for the research that may be of interest to your readers.

 We hope that you will find the revised paper suitable for publication, and we look forward to contributing to your journal. Please do not hesitate to contact us with other questions or concerns regarding the manuscript.

Best regards,

 

Reviewer #1 

In the abstract the authors write in the first sentence that the initial and mean rSO2 play a role in the efficiency of CPR. However it is impossible that an initial value as also the mean values, which is generated posthoc can influence the quality of CPR. The authors should adapt this sentence.

Response: We are sorry for this. We meant "regional cerebral oxygen saturation (rSO2)".

What do the authors mean with patients status in the abstract and further in the manuscript? In the Tables it is the location of the cardiac arrest. The authors should indicate this in the abstract and the first time they mention this in the manuscript.

Response: The Reviewer is right. It was corrected accordingly.

The authors make a difference between OHCA and IHCA patients however they should also investigate the moment of measurements. As in some studies, cerebral oxygenation is measured pre-hospital, sometimes at the ED of OHCA, sometimes In hospital at the place of arrest or at the ED once the IHCA patient is brought to the ED. You can expect differences in saturation values as the time between cardiac arrest and start of measurements is different. Do the authors analyze the data categorized following the moment of measurement OH or IH?

Response: We thank the Reviewer for the comment. As this was a meta-analysis, we had to work with the data available from the previous papers. As such, unfortunately, we could not perform the analysis suggested by the Reviewer.

The authors write that cardiac arrest is a disease however they should adapt this as it is an acute endpoint of many diseases.

Response: We agree with the Reviewer. It was edited as suggested.

Line 41: the authors give a survival rate of 6,3% however they don't give any reference. Can they add a reference?

Response: We thank the reviewer for this kind reminding, a reference was added.

Line 52: this sentence should be adapted as none of the other studies until today could associate rSO2 and neurological outcome.

Response: We agree with the Reviewer. It was edited.

Line 53: rSO2 is not widely used pre-operative.

Response: We agree with the Reviewer. It was edited.

Line 59: what do they mean with specific characteristics?

Response: We thank the Reviewer. We meant patients with different characteristics. It was corrected.

The authors should also search under near infra red spectroscopy.

Response: We thank the Reviewer. This keyword was not included in our original protocol. Nevertheless, we made verification, and it did not yield additional eligible studies.

How and why was decided to use initial and mean rSO2 values?

Response: The present study was a meta-analysis of published studies. We used the data available from those studies.

Are the authors sure they included all manuscript about cerebral saturation and cardiac arrest during they research period? A recent study was published by Genbrugge et al. using cerebral oximetry in CA patients. Why was this study excluded? Why are the studies of Ito et al. excluded?

Response: We thank the Reviewer for the comment. We performed the search in May 2018. The study by Genbrugge et al. was published in August 2018. The studies by Ito et al. were not included because they did not match the inclusion criteria.

On what is the subgroup analysis based?

Response: We thank the Reviewer for the comment. Subgroup analyses are often used in meta-analyses in order to determine whether some factors (e.g., publication year, study type, characteristics of the patients, etc.) could influence the results.

Did the authors also investigated the different devices used to measure rSO2? As some devices measure until 0 other have a higher cut off point. This can influence in the calculated mean values and initial values as some devices for example indicate 15 for all values below 15

Response: We thank the Reviewer for the comment. Unfortunately, the number of different devices was too large, and the number of studies for some devices was too small for a reliable analysis. This will have to be examined in a future study with a different study design to try to include as many papers as possible for each device. This was included as a limitation.

Reviewer #2

-All results should be separated based on prospective vs. retrospective and in-hospital vs out-of-hospital throughout the entire paper as it is not ideal to perform statistics combining these groups

Response: We agree with the Reviewer that combining the two types of studies is not ideal, but it is a limitation of many meta-analyses. Nevertheless, the results of the meta-regression (Table 3) show that the study type did not influence the outcome of the meta-analysis. Perhaps when additional studies have been published, we will be able to perform a meta-analysis specifically for prospective and retrospective studies. This is already included in the Limitations.

-In your methods, describe what the term 'patient status' means, as this is a very general term

Response: We thank the Reviewer. We meant "the location of the cardiac arrest". It was clarified throughout the manuscript.

-More detail is needed in the statistical analysis section to describe what you mean by 'subgroup analysis' and 'meta-regression analysis' as these terms alone don't convey what you are specifically addressing with these analyses (i.e. what outcomes are you looking at)

Response: We thank the Reviewer. Subgroup analyses and meta-regression were performed to determine whether the characteristics of the studies and populations of patients influenced the conclusion of the meta-analysis [1]. The subgroup analyses were based on the publication year, study design, mean age, the percentage of males, the location of cardiac arrest, and study quality.

-Instead of saying the relationships can be biased by 'percentage of males', it would be better to state they are influenced by gender. Additionally, if gender does influence the results, it would be beneficial to add a table that specifically separates results for males and females if this data is available.

Response: We thank the Reviewer. The statement was edited accordingly. As for the data, we could not extract the separate data by sex from the included studies.

References

1. Thompson SG, Higgins JP (2002) How should meta-regression analyses be undertaken and interpreted? Stat Med 21: 1559-1573.

---

## [Decision Letter · Decision Letter 1]

13 Mar 2020

PONE-D-19-30622R1

Association between cerebral oximetry and return of spontaneous ci rculation following cardiac arrest: a systematic review and meta-analysis

PLOS ONE

Dear Dr Zhang,

Thank you for submitting your manuscript to PLOS ONE. After careful consideration, we feel that it has merit but does not fully meet PLOS ONE’s publication criteria as it currently stands. Therefore, we invite you to submit a revised version of the manuscript that addresses the points raised during the review process.

We would appreciate receiving your revised manuscript by April 30th. To enhance the reproducibility of your results, we recommend that if applicable you deposit your laboratory protocols in protocols.io, where a protocol can be assigned its own identifier (DOI) such that it can be cited independently in the future. For instructions see: http://journals.plos.org/plosone/s/submission-guidelines#loc-laboratory-protocols

We look forward to receiving your revised manuscript.

Kind regards,

Steve Lin

Academic Editor

PLOS ONE

Reviewers' comments:

Reviewer's Responses to Questions

**Comments to the Author**

1. If the authors have adequately addressed your comments raised in a previous round of review and you feel that this manuscript is now acceptable for publication, you may indicate that here to bypass the “Comments to the Author” section, enter your conflict of interest statement in the “Confidential to Editor” section, and submit your "Accept" recommendation.

Reviewer #1: (No Response)

Reviewer #2: All comments have been addressed

2. Is the manuscript technically sound, and do the data support the conclusions?

Reviewer #1: Yes

Reviewer #2: Yes

3. Has the statistical analysis been performed appropriately and rigorously? 

Reviewer #1: Yes

Reviewer #2: Yes

4. Have the authors made all data underlying the findings in their manuscript fully available?

Reviewer #1: Yes

Reviewer #2: Yes

5. Is the manuscript presented in an intelligible fashion and written in standard English?

Reviewer #1: Yes

Reviewer #2: Yes

6. Review Comments to the Author

Reviewer #1: Thank you for taking the time to submit revisions. Howver some facts are still not clear.

Sentence 169: a word is missing

How was decided to use publication year, study type,...in the subgroup analysis? What do the authors mean with characteristics of the patient? Can they be more precise?

The study by Genbrugge et al. was published online in March (see Pubmed)

Why do the authors not use mean rSO2 during the last 5mins? Increase of rSO2 during CPR? Mean of the first min of rSO2 in their analysis? The answer provided we used the data available from those studies insufficient as some studies do present these data. How did the authors decided not to includ these data? They should mention this in the limitations of this study?

Reviewer #2: The authors have adequately addressed my original concerns.

1) There are just a few grammatical issues to address:

-Lines 58-61: Please break into 2 sentences, for example: "A previous meta-analysis based on 9 studies found that the return of spontaneous circulation (ROSC) in CA patients is associated with high initial and average rSO2. Whether or not these correlations varied among patients with different characteristics are not yet illustrated

-Lines 80-81: Please add the word "and" in between "search" and "selected"

-Lines 186: Please change the word "was" to "is"

-Line 244: Please change the word "indicated" to "suggest" as it is overall a fairly weak correlation with low quality of evidence

7. PLOS authors have the option to publish the peer review history of their article (what does this mean?). If published, this will include your full peer review and any attached files.

Reviewer #1: No

Reviewer #2: No

---

## [Author Response · Author response to Decision Letter 1]

29 Mar 2020

Manuscript ID: PONE-D-19-30622R1

Title: Association between cerebral oximetry and return of spontaneous circulation following cardiac arrest: a systematic review and meta-analysis

Journal: PLOS ONE

Response to Reviewers' comments

Dear Editor, 

 We thank you for your careful consideration of our manuscript. We appreciate your response and overall positive feedback and made modifications to improve the manuscript. 

 We hope that you will find the revised paper suitable for publication, and we look forward to contributing to your journal. Please do not hesitate to contact us with other questions or concerns regarding the manuscript.

Best regards,

 

Reviewer #1 

Sentence 169: a word is missing

Response: We thank the Reviewer for the comment. It was corrected, and the manuscript was proofread.

How was decided to use publication year, study type,...in the subgroup analysis? What do the authors mean with characteristics of the patient? Can they be more precise?

Response: We thank the Reviewer for the comment. We added to the Methods: “The subgroup analyses were based on the publication year (this could be a confounder because of the changes in practice and guidelines over time), study design (prospective trials have more power than retrospective studies), mean age (age might influence cerebral oximetry and the outcomes of cardiac arrest), sex (males usually have worst cardiac outcomes than females), the location of cardiac arrest (cardiac arrests occurring at the hospital will be managed more promptly than those occurring outside), and study quality (study and data quality might affect the quality of the analyses and conclusions).” The patient characteristics were clarified.

The study by Genbrugge et al. was published online in March (see Pubmed)

Response: We thank the Reviewer for the comment. We added this study to the discussion, but it was included in the analyses since it was published outside the study period.

Why do the authors not use mean rSO2 during the last 5mins? Increase of rSO2 during CPR? Mean of the first min of rSO2 in their analysis? The answer provided we used the data available from those studies insufficient as some studies do present these data. How did the authors decided not to includ these data? They should mention this in the limitations of this study?

Response: We thank the Reviewer for the comment. Not all studies report the same outcomes, and the reported data have to be analyzable in the context of a meta-analysis. Therefore, we had to select the variables that were the most consistently reported among the included studies and had clinical meaning. This was added as a limitation.

Reviewer #2

1) There are just a few grammatical issues to address:

-Lines 58-61: Please break into 2 sentences, for example: "A previous meta-analysis based on 9 studies found that the return of spontaneous circulation (ROSC) in CA patients is associated with high initial and average rSO2. Whether or not these correlations varied among patients with different characteristics are not yet illustrated

-Lines 80-81: Please add the word "and" in between "search" and "selected"

-Lines 186: Please change the word "was" to "is"

-Line 244: Please change the word "indicated" to "suggest" as it is overall a fairly weak correlation with low quality of evidence

Response: We thank the Reviewer for the comment. Those corrections were made and the manuscript was proofread.

---

## [Decision Letter · Decision Letter 2]

23 Apr 2020

PONE-D-19-30622R2

Association between cerebral oximetry and return of spontaneous ci rculation following cardiac arrest: a systematic review and meta-analysis

PLOS ONE

Dear Dr Liu,

Thank you for submitting your manuscript to PLOS ONE and making the necessary revisions for publication. There is one minor comment and suggestion from a reviewer that we ask you to address before your manuscript can be accepted for publication.

We would appreciate receiving your revised manuscript by May 8, 2020. To enhance the reproducibility of your results, we recommend that if applicable you deposit your laboratory protocols in protocols.io, where a protocol can be assigned its own identifier (DOI) such that it can be cited independently in the future. For instructions see: http://journals.plos.org/plosone/s/submission-guidelines#loc-laboratory-protocols

We look forward to receiving your revised manuscript.

Kind regards,

Steve Lin

Academic Editor

PLOS ONE

Reviewers' comments:

Reviewer's Responses to Questions

**Comments to the Author**

1. If the authors have adequately addressed your comments raised in a previous round of review and you feel that this manuscript is now acceptable for publication, you may indicate that here to bypass the “Comments to the Author” section, enter your conflict of interest statement in the “Confidential to Editor” section, and submit your "Accept" recommendation.

Reviewer #1: (No Response)

2. Is the manuscript technically sound, and do the data support the conclusions?

Reviewer #1: Yes

3. Has the statistical analysis been performed appropriately and rigorously? 

Reviewer #1: Yes

4. Have the authors made all data underlying the findings in their manuscript fully available?

Reviewer #1: Yes

5. Is the manuscript presented in an intelligible fashion and written in standard English?

Reviewer #1: Yes

6. Review Comments to the Author

Reviewer #1: Thank you for your answers. I still have one question.

In line 213 and further the authors added two sentences using a reference that is not about cardiac arrest patients. They should delete these sentences as they don't have have an added value for the discussion.

The authors replied that they added the study of Genbrugge et al. in the discussion and not in the analyses as it was published outside the study period. However the study period was until may 2018 and the manuscript was published in March 2018. As the number of included patients in that study is quiet big I think this is a methodological mistake of the authors if it was not included initially and I understand that it is a lot of work to redo the analyses. However at least they can add this to the disccusion as also the manuscript. The authors replied they added this manuscript to the disccusion however I don't find this. So can they pleas adjust this?

(Resuscitation. 2018 Aug;129:107-113. doi: 10.1016/j.resuscitation.2018.03.031. Epub 2018 MAR23 (!!!!) .Cerebral saturation in cardiac arrest patients measured with near-infrared technology during pre-hospital advanced life support. Results from Copernicus I cohort study. Genbrugge C1, De Deyne C2, Eertmans W3, Anseeuw K4, Voet D5, Mertens I6, Sabbe M7, Stroobants J8, Bruckers L9, Mesotten D10, Jans F11, Boer W12, Dens J13.)

7. PLOS authors have the option to publish the peer review history of their article (what does this mean?). If published, this will include your full peer review and any attached files.

Reviewer #1: No

---

## [Author Response · Author response to Decision Letter 2]

11 May 2020

Manuscript ID: PONE-D-19-30622R3

Title: Association between cerebral oximetry and return of spontaneous circulation following cardiac arrest: a systematic review and meta-analysis

Journal: PLOS ONE

Response to Reviewers' comments

Dear Editor,

 We thank you for your careful consideration of our manuscript. We appreciate your response and overall positive feedback and made modifications to improve the manuscript. 

 We hope that you will find the revised paper suitable for publication, and we look forward to contributing to your journal. Please do not hesitate to contact us with other questions or concerns regarding the manuscript.

Best regards,

 

Reviewer #1 

In line 213 and further the authors added two sentences using a reference that is not about cardiac arrest patients. They should delete these sentences as they don't have have an added value for the discussion.

 Response: We thank the Reviewer for the comment. Those two sentences were deleted.

The authors replied that they added the study of Genbrugge et al. in the discussion and not in the analyses as it was published outside the study period. However the study period was until may 2018 and the manuscript was published in March 2018. As the number of included patients in that study is quiet big I think this is a methodological mistake of the authors if it was not included initially and I understand that it is a lot of work to redo the analyses. However at least they can add this to the disccusion as also the manuscript. The authors replied they added this manuscript to the disccusion however I don't find this. So can they pleas adjust this?

(Resuscitation. 2018 Aug;129:107-113. doi: 10.1016/j.resuscitation.2018.03.031. Epub 2018 MAR23 (!!!!) .Cerebral saturation in cardiac arrest patients measured with near-infrared technology during pre-hospital advanced life support. Results from Copernicus I cohort study. Genbrugge C1, De Deyne C2, Eertmans W3, Anseeuw K4, Voet D5, Mertens I6, Sabbe M7, Stroobants J8, Bruckers L9, Mesotten D10, Jans F11, Boer W12, Dens J13.)

 Response: Again, we thank the Reviewer for the comment. We agree that the epub date is before May 2018, but the actual publication was in August 2018. Because access to the Internet is not as free and easy in China as it can be in other countries, there might be a lag in access to the latest data, and we might have to wait for the paper versions, in some cases. We agree that it would add much to our meta-analysis in terms of patient numbers, but we added it in the Discussion.

---

## [Decision Letter · Decision Letter 3]

8 Jun 2020

Association between cerebral oximetry and return of spontaneous circulation following cardiac arrest: a systematic review and meta-analysis

PONE-D-19-30622R3

We’re pleased to inform you that your manuscript has been judged scientifically suitable for publication and will be formally accepted for publication once it meets all outstanding technical requirements.

Kind regards,

Steve Lin

Academic Editor

PLOS ONE

Additional Editor Comments (optional):

Reviewers' comments:

Reviewer's Responses to Questions

**Comments to the Author**

1. If the authors have adequately addressed your comments raised in a previous round of review and you feel that this manuscript is now acceptable for publication, you may indicate that here to bypass the “Comments to the Author” section, enter your conflict of interest statement in the “Confidential to Editor” section, and submit your "Accept" recommendation.

Reviewer #1: All comments have been addressed

2. Is the manuscript technically sound, and do the data support the conclusions?

Reviewer #1: Yes

3. Has the statistical analysis been performed appropriately and rigorously? 

Reviewer #1: Yes

4. Have the authors made all data underlying the findings in their manuscript fully available?

Reviewer #1: Yes

5. Is the manuscript presented in an intelligible fashion and written in standard English?

Reviewer #1: Yes

6. Review Comments to the Author

Reviewer #1: The authors addressed all my concerns however in a next manuscript they should verify if they included all published articles and if not they should justify in the limitation or discussion why not.

7. PLOS authors have the option to publish the peer review history of their article (what does this mean?). If published, this will include your full peer review and any attached files.

Reviewer #1: No

---

## [Editor Report · Acceptance letter]

19 Aug 2020

PONE-D-19-30622R3 

Association between cerebral oximetry and return of spontaneous circulation following cardiac arrest: a systematic review and meta-analysis 

Dear Dr. Liu:

I'm pleased to inform you that your manuscript has been deemed suitable for publication in PLOS ONE. Congratulations! Your manuscript is now with our production department. 

Kind regards, 

on behalf of

Dr. Steve Lin 

Academic Editor

PLOS ONE